# Bacterial dormancy: A subpopulation of viable but non-culturable cells demonstrates better fitness for revival

Sariqa Wagley[1]*, Helen Morcrette[1], Andrea Kovacs-Simon[1], Zheng R. Yang[1], Ann Power[2], Richard K. Tennant[2], John Love[2], Neil Murray[3], Richard W. Titball[1], Clive S. Butler[1]*

1 Biosciences, College of Life and Environmental Sciences, University of Exeter, Exeter, United Kingdom,
2 BioEconomy Centre, The Henry Wellcome Building for BioCatalysis, Biosciences, Exeter, United Kingdom,
3 Lyons Seafoods, Fairfield House, Warminster, Wiltshire, United Kingdom

* s.wagley@exeter.ac.uk (SW); c.s.butler@exeter.ac.uk (CSB)

**Data Availability Statement:** The data is all found in this manuscript.

**Funding:** Work is funded by Biotechnology, and Biological Research Council (BBSRC) reference BB/

## Abstract

The viable but non culturable (VBNC) state is a condition in which bacterial cells are viable and metabolically active, but resistant to cultivation using a routine growth medium. We investigated the ability of *V. parahaemolyticus* to form VBNC cells, and to subsequently become resuscitated. The ability to control VBNC cell formation in the laboratory allowed us to selectively isolate VBNC cells using fluorescence activated cell sorting, and to differentiate subpopulations based on their metabolic activity, cell shape and the ability to cause disease in *Galleria mellonella*. Our results showed that two subpopulations (P1 and P2) of *V. parahaemolyticus* VBNC cells exist and can remain dormant in the VBNC state for long periods. VBNC subpopulation P2, had a better fitness for survival under stressful conditions and showed 100% revival under favourable conditions. Proteomic analysis of these subpopulations (at two different time points: 12 days (T12) and 50 days (T50) post VBNC) revealed that the proteome of P2 was more similar to that of the starting microcosm culture (T0) than the proteome of P1. Proteins that were significantly up or down-regulated between the different VBNC populations were identified and differentially regulated proteins were assigned into 23 functional groups, the majority being assigned to metabolism functional categories. A lactate dehydrogenase (lldD) protein, responsible for converting lactate to pyruvate, was significantly upregulated in all subpopulations of VBNC cells. Deletion of the lactate dehydrogenase (RIMD2210633:Δ*lldD*) gene caused cells to enter the VBNC state significantly more quickly compared to the wild-type, and adding lactate to VBNC cells aided their resuscitation and extended the resuscitation window. Addition of pyruvate to the RIMD2210633:Δ*lldD* strain restored the wild-type VBNC formation profile. This study suggests that lactate dehydrogenase may play a role in regulating the VBNC state.

## Author summary

Members of the Proteobacteria are reported to adopt a survival strategy and enter a viable but non culturable (VBNC) state, when exposed to stressful or non-permissible growth

N016513/1. SW, CSB, RWT received the funding. https://bbsrc.ukri.org/ The funders had no role in study design, data collection and analysis, decision to publish, or preparation of the manuscript.

**Competing interests:** The authors have declared that no competing interests exist.

conditions. This is a characteristic employed widely in the natural environment in order for bacteria to survive harsh environmental conditions over a long period. In spite of the importance of the VBNC state in microbiology, we know little about the molecular makeup of VBNC cells. In this study, using the model organism *Vibrio parahaemolyticus*, we resolved that distinct subpopulations of bacteria exist in the VBNC state and these have different resuscitation potentials and distinct proteomic profiles. We also discovered that deletion of the gene encoding the enzyme lactate dehydrogenase (lldD) triggered the cells to enter the VBNC state, and adding lactate to VBNC cells extended their resuscitation potential window. The ability for bacteria to survive in the VBNC state might be linked to their ability to overcome oxidative stress.

## Introduction

Members of the Proteobacteria are reported to have the ability to form viable but non-culturable (VBNC) cells, a fundamental survival mechanism that allows bacteria to 'hibernate' or lay dormant until conditions become more favourable to support their growth [1,2]. VBNC bacteria can continue to utilise nutrients, retain their plasmids, undergo cell rounding and retain virulence properties [3–10]. Importantly, VBNC bacteria remain metabolically active albeit at a reduced capacity, but no longer form colonies on standard culture media [11–13]. Following environmental stimuli or permissible growth conditions, some VBNC cells can 'resuscitate', restoring their ability to grow on media [1,13]. The VBNC state has been well documented in *Vibrio* species and *V. parahaemolyticus* is a good model organism for studying VBNC cells for a number of reasons [14–16]. Firstly, the VBNC state can be induced in *V. parahaemolyticus* by low temperatures and salinity, secondly *V. parahaemolyticus* VBNC cells can be resuscitated by increasing the temperature in the medium and thirdly, the period of resuscitation of VBNC cells is well documented to be approximately 2 weeks after cells have become unculturable [8,11,15–18].

*V. parahaemolyticus* is also of particular interest because it is the leading cause of seafood associated gastroenteritis and is abundant in shellfish when sea temperatures exceed 18˚C, coinciding with elevated disease burden. In the absence of ideal growth conditions, culturable *V. parahaemolyticus* can no longer be detected in shellfish samples and it is thought that *Vibrio* species do not survive well at low temperatures. Previous studies have shown that *V. parahaemolyticus* can appear seasonally in the environment [19], indicating a possible dormant state for these bacteria during colder temperatures. *V. parahaemolyticus* in the VBNC state may constitute a reservoir of bacteria that can be reactivated later under more favourable conditions. Understanding the relationship between VBNC cells and cells that are able to grow is critical for understanding the incidence of disease potential from the environment.

To date, little is understood about the genetic control underlying the VBNC state and regulators such as RpoS and OxyR have been identified in bacteria as playing a role in VBNC formation [1,20]. The lack of the stress regulator RpoS has been shown to decrease the ability of *E. coli* to remain in the VBNC state [21,22] and in *V. parahaemolyticus* repression of *rpoS* expression was observed when cells could no longer be resuscitated at 37˚C [23]. In *V. vulnificus* the oxidative stress regulator OxyR has been shown to regulate the activity of catalase, which was required to degrade hydrogen peroxide generated in response to cold shock [20]. More recently, the gene *cpdA* was shown to be involved by regulating cAMP- receptor proteins in the VBNC cell environment and a lack of cAMP-CRP retained colony formation in *E. coli* VBNC induced conditions [24]. Other studies have shown that *E. coli* VBNC cells are identical

to persister cells and that control of persister cells are due to ribosome dimerization and the use of chemotaxis to acquire nutrients [25,26].

Transcriptomic profiling of VBNC cells has been used to identify the genetic determinants of the VBNC state and a recent study demonstrated differential gene expression in *V. parahaemolyticus* VBNC cells compared to exponential or stationary phase cells [27]. The study revealed that genes involved in glutamate synthesis, biofilm maintenance, DNA repair and transportation were up regulated at least 4-fold during the VBNC state. Collectively, the transcriptome studies on VBNC *Vibrio* cells [27,28] are useful but, crucially, do not identify genes where the function is unknown that may play a role in VBNC formation.

Investigating the proteome of cells in the VBNC state and comparing it to the proteome from growing culturable cells, is an augmented and complementary approach and provides valuable information regarding differentially expressed proteins and enzymes. Two recent studies have used a proteomic approach to investigate *V. parahaemolyticus* in the VBNC and resuscitation states [29,30]. The analysis reveals that when compared to exponentially growing cells, 36 proteins were significantly down regulated and 15 were significantly up regulated as the cell population entered VBNC. The majority of regulated proteins were found to be associated with; translation, structural constituent of ribosome, rRNA binding, siderophore transmembrane transporter activity, receptor activity and bacterial-type flagellum organization [29]. Upon resuscitation from the VBNC state, 429 proteins were found to be differentially expressed, with 330 significantly up-regulated and 99 down-regulated [30]. Whilst these studies have provided a general overview of protein expression in *V. parahaemolyticus* upon entry and exit from the VBNC state, they have considered the culture as a homogeneous population of cells that all display a similar response. However, SEM imaging has previously indicated that *V. parahaemolyticus* VBNC cells actually form a heterogeneous population; where small coccoid cells and flattened larger cells coexist [18]. The contribution of these morphologically differentiated cells to a global VBNC proteome is currently unknown, but is absolutely crucial for a more accurate, mechanistic understanding of bacterial survival and subsequent growth.

In this investigation, we used imaging flow cytometry (IFC) and fluorescence activated cell sorting (FACS) to characterise and isolate two morphologically distinct subpopulations (P1 and P2) of *V. parahaemolyticus*, as they enter the VBNC state. Cells from either subpopulation can remain dormant in the VBNC state for long periods, but the P1 and P2 subpopulations displayed different abilities for revival under favourable conditions. To understand the basis of these differences, the proteomes of P1 and P2 were analysed and shown to possess distinct features. The majority of the significantly upregulated proteins belonged to metabolic functional categories and included a lactate dehydrogenase (LldD; VPA1499) one of the most highly abundant proteins in both P1 and P2 subpopulations. We demonstrated that deletion of the lactate dehydrogenase encoding gene caused the cells to enter the VBNC state significantly quicker than the wild-type; and that supplementation with lactate aided resuscitation and extended the resuscitation window indicating a role of lactate dehydrogenase in the VBNC state. This work advances our understanding of the VBNC consortium and provides novel molecular insight to VBNC subpopulation morphologies and their virulence potential.

## Results

### Induction of cells into the VBNC state depends on inoculum age

*V. parahaemolyticus* strain RIMD2210633 was used to establish high cell density ($10^9$ CFU/ml) laboratory microcosms. Initial experiments showed that we could impose conditions in which cells entered a VBNC state (hereafter referred to as microcosms). We could resuscitate cells to

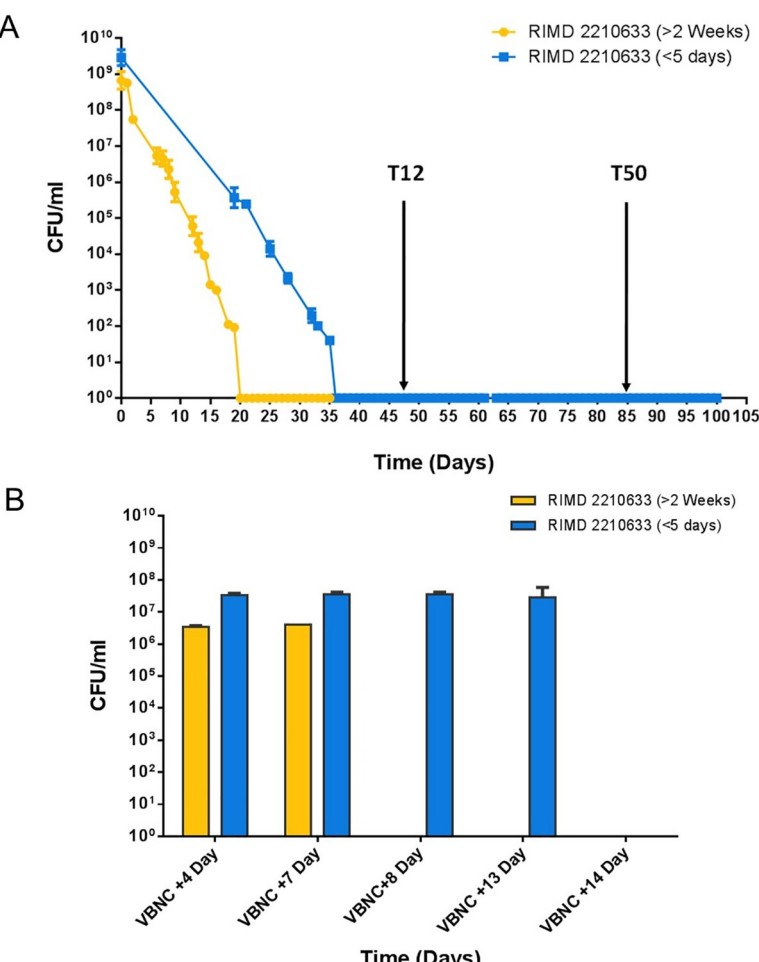

**Fig 1. The age of the strain can affect VBNC formation and resuscitation.** A) The number of culturable cells of *V. parahaemolyticus* RIMD 2210633 in the microcosms over time using either fresh cultures that were ≤5 days out of the freezer or older cultures that had been on agar plates for ≥14 days. When older cultures were used to set up microcosms it took ~20 days for cells to reach unculturable while microcosms prepared with cultures that were less than 5 days old from freezer stocks took longer to become unculturable in the microcosm. The detection limit of CFU was 0.2 cells /ml. B) Resuscitation of cells was tested when all the cells in the population had turned unculturable. When older cultures were used to set up a microcosm a resuscitation window of 7 days was observed while a resuscitation window of 2 weeks could be observed in microcosms set up with fresher cultures.

culturable forms depending on; the age of the culture, the handling of cells to avoid damage during preparation and the methods used to induce VBNC formation and resuscitate the cells (S1 Text).

When 5-day old cultures were used to establish VBNC microcosms, the number of culturable cells declined to undetectable levels within 30–35 days (Fig 1A). In VBNC microcosms set up using cultures that had been stored on agar for at least 2 weeks, the number of culturable cells declined more rapidly, taking approximately 20 days to reach undetectable levels (Fig 1A). We found that 0.63% of cells could be resuscitated for up to 7 days when the VBNC microcosms were prepared from >14 day cultures and 1.3% of cells could be resuscitated for up to 13 days when 5 day cultures were used to establish the microcosm (Fig 1B). For subsequent experiments we used VBNC microcosms established from cultures that were consistently less than 5 days old (Fig 1A).

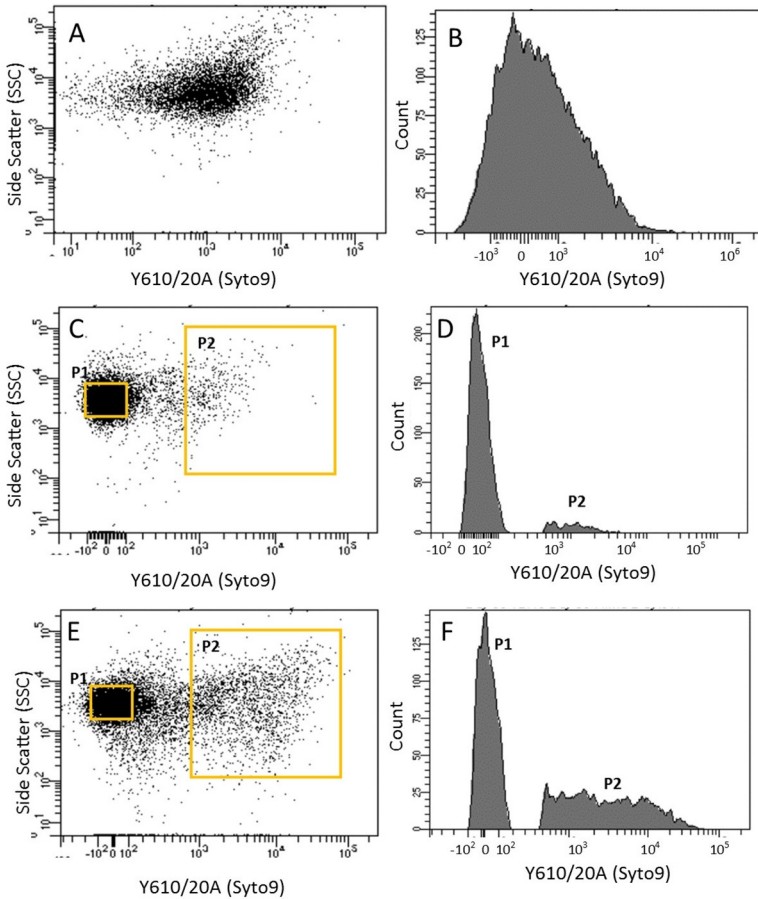

**Fig 2. Flow cytometry analysis of microcosms.** Dot plots (Left) and corresponding histograms (Right) for Time point T0 (A-B), Time point T12 (C-D) and Time point T50 (E-F). Left: Dot plot of side scatter area vs. Y610/20 emission area management. Right: Line gate was used to select population P1 and P2 at T12 and T50 and plotted on a histogram of Y610/20 emission area to highlight proportions of cells in populations P1 and P2. Populations P1 and P2 became visible on dot plots and histograms once the cells in the microcosm population had turned unculturable. Intensity of fluorescence increased (as measured on the YG610/20 laser) and a peak on Y610/20 was seen which corresponded to a large side scatter.

## Flow cytometry analysis of VBNC cells

Flow cytometry was used to analyse 10,000 cells/events from the microcosm at 3 different time points; T0 (start of the microcosm), T12 (12 days post VBNC formation) and T50 (50 days post VBNC formation) (Fig 1A). Fig 2A and 2B shows that at T0 over 40% (±10%) of the cells stained as live and metabolically active, while approximately 45% (±15%) of cells stained as damaged or dead. Analysis of the microcosm at T12 using FACS after PI staining showed that damaged/dead cells were not visible, indicating that the dead/damaged cells seen at T0 had repaired, lysed or degraded by this stage. Conversely, all of the cells that stained alive and with an intact membrane, at T12 and T50 fell into two distinct populations based on their size and fluorescence signals (Fig 2C and 2D). These two populations were designated as Population 1 (P1), in which cells had a low fluorescence signal and were a smaller size, and Population 2 (P2) in which the cells had a higher fluorescence signal and were larger. The percentage of cells moving from gates for population P1 changed from 1.27% at T0 to 39.8% or 27% at T12 or T50 respectively (Fig 2C and 2E). While for population P2 the percentage of cells changed from 21% at T0 down to 0.1% or 6.4% at T12 or T50 respectively (Fig 2C and 2E).

**Table 1. Different VBNC subpopulations and morphologies of cells identified by IFC analysis compared to T0 population of cells. Cell lengths and cell widths are the size with standard deviation.**

| Population | Morphology | Number of cells | Fraction of VBNC population | Cell length (μm) | Cell width (μm) |
|---|---|---|---|---|---|
| T0 | Small rods | 10992 | - | 1.26 ± 0.1 | 1.00 ± 0.04 |
| P1 | Small coccoid | 8434 | 89.4% | 1.31 ± 0.18 | 1.07 ± 0.17 |
| P2 | Large coccoid | 950 | 10.07% | 6.34 ± 1.10 | 4.30 ± 0.48 |
| P2 | Filaments | 50 | 0.53% | 2.69 ± 0.60 | 1.51 ± 0.45 |

Cell lengths and cell widths are the size with standard deviation.

## Morphological analysis of the P1 and P2 subpopulations

Next, the physiological differences between T0 cells, and VBNC subpopulations P1 and P2 at time points T12 or T50 were investigated. Using Imaging Flow Cytometry (IFC) and FACS we identified culturable *V. parahaemolyticus* cells in the T0 population as 1.26 ± 0.1 μm in length and 1.00 ± 0.04 μm in width (Table 1 and Fig 3A). Using IFC we analysed ~1900–5000 events (*n* = 3 ± SD) from the VBNC microcosms and found morphological differences between the P1 and P2 populations. P1 cells were a similar size to T0 cells, measuring 1.31 ± 0.18 μm in length and 1.07 ± 0.17 μm in width (Fig 3B). These P1 cells accounted for 89% of the sample. Subpopulation P2 consisted of filaments or chains of rods that accounted for 0.5% of the microcosm and had an average length 2.69 ± 0.6 μm and an average width of 1.51 ± 0.5 μm (Fig 3C). Subpopulation P2 also contained large coccoid cells that were 6.34 ± 1.1 μm in length and 4.3 ± 0.48 μm in width and accounted for 10% of the microcosm population (Fig 3D).

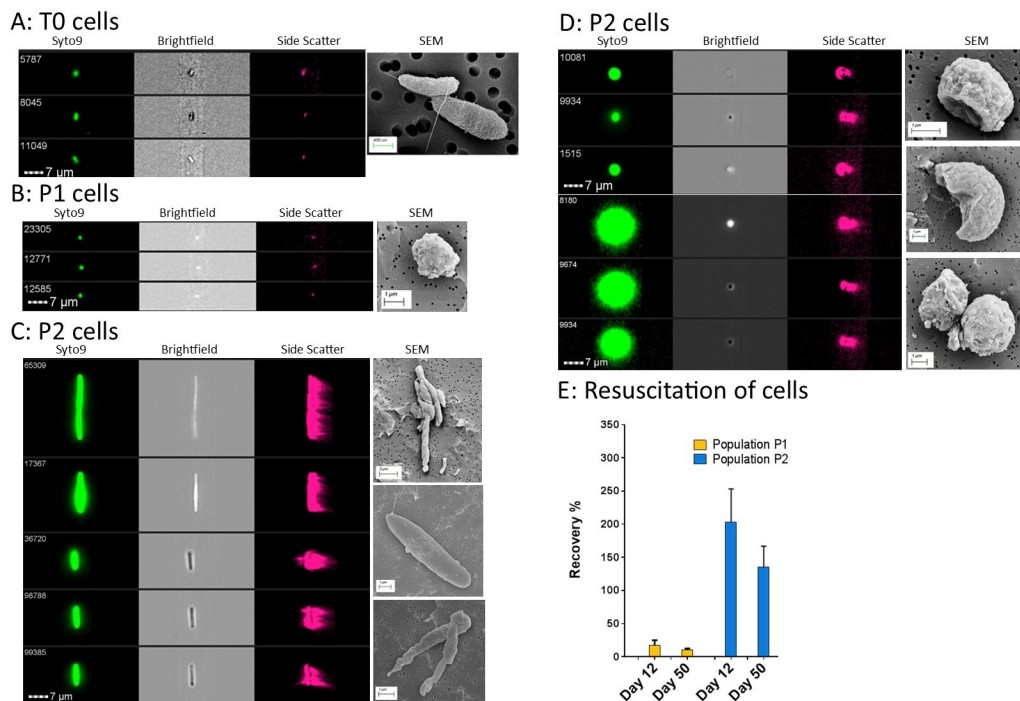

**Fig 3. Analysis of the cell morphology in different VBNC subpopulations.** Cells analysed using the ImageStream Technology were stained with Syto9 stain before imaging. Examples of healthy rod shaped *V. parahaemolyticus* at T0 are shown in panel A, small coccoid VBNC cells from population P1 in panel B, large rods/filaments and large coccoid cells from population P2 in panel C and D respectively. Images are accompanied with representative SEM pictures. Panel E shows the percentage recovery (resuscitation) of VBNC cells stained with Syto9 in subpopulations P1 and P2. Data is from time point T12 and T50 and representative of 4 microcosms and standard deviation is shown ± SD.

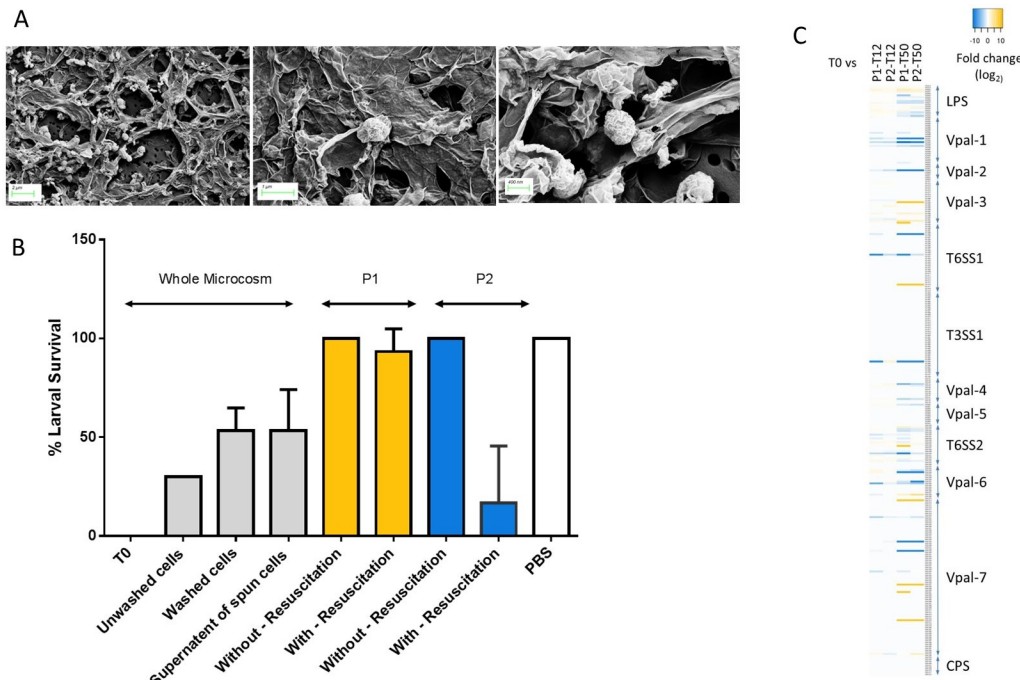

**Fig 4. Virulence potential of *V. parahaemolyticus* RIMD2210633 cell types.** Panel A shows SEM pictures of VBNC cells with extracellular matrix attached surrounding the cells. Panel B shows survival of *Galleria mellonella* after 48 h when injected with different VBNC cell types. Panel C shows a heat map identifying regulation among known virulence proteins.

Scanning electron microscopy (SEM) images were acquired of the whole microcosm and FACS sorted VBNC subpopulations P1 and P2, to verify the morphological observations seen by IFC. SEM analysis of whole VBNC microcosms showed that VBNC cells were part of a complex extracellular matrix of which VBNC cells were embedded or attached (Fig 4A). SEM images of cells in VBNC subpopulation P1 were seen as smaller coccoid cells while VBNC subpopulation P2 were verified as larger dense coccoid cells that were sometimes in doublets. SEM images also captured long chains of hollow cells in the microcosm in VBNC subpopulation P2. Transmission electron microscopy was also carried out on FACS sorted VBNC subpopulations P1 and P2. Subpopulation P1 cells showed contents of the cytosol was still present and the cell wall was extended and sometimes bulging with empty spaces. Subpopulation P2 showed cells to have intact cell membranes and empty spaces in the cytosol (S1 Fig).

## Subpopulations of VBNC cells have different resuscitation capabilities

Next, the entire VBNC microcosm was tested at time points T12 and T50 for its ability to be resuscitated. When T12 resusciated cells were incubated in PBS and subjected to an increase in temperature they could be resuscitated and had a "return to growth" phenotype. This is consistent with published data that suggests that *V. parahaemolyticus* VBNC cells can be resuscitated for approximately 2 weeks after their formation [15–17]. At T50 VBNC cells could not be resuscitated using this method.

To test resuscitation capabilities of individual VBNC subpopulations we used FACS to collect 250,000 or 50,000 cells/events from the P1 and P2 populations respectively at time point T12 and T50. We have presented the number of resuscitated cells as a percentage of the total number collected. The cells in subpopulation P2 were mainly large coccoid in shape and could

be resuscitated at either T12 or T50 (Fig 3E). A recovery of over 100% from some samples indicates that some cells in this subpopulation were in doublets, chains, or in large coccoid cells consisting of more than one cell. Approximately 14% of T12/T50 VBNC cells in population P1 could be resuscitated. When resuscitated the coccoid cells returned to rod shaped bacteria. These results indicate that particular subpopulations of *V. parahaemolyticus* can remain in the VBNC state for long periods and can be resuscitated more than 2 weeks after their formation.

## Virulence in *G. mellonella* larvae

We have shown previously that *Galleria mellonella* (wax moth) larvae can be used to assess virulence of *V. parahaemolyticus* [31], including strain RIMD2210633 and the median lethal dose of this strain was approximately 100 CFU [31]. When we dosed larvae with approximately $10^5$ CFU of T0 cells, ~50% of the larvae succumbed to infection (Fig 4B). Next, we assessed the virulence of VBNC cells at time point T12 in the whole microcosms. Using basic resuscitation methods, we calculated that approximately $10^5$ CFU were injected into larvae and found that 70% of the larvae had died by 48 h. Subsequently, culturable *V. parahaemolyticus* could not be isolated from the larvae. In order to demonstrate that virulence was due to the VBNC cells and not solely due to the accumulation of a secreted toxin from T0 cells, VBNC cells from T12 were washed and injected into the larvae. After 48hrs, ~50% of the larvae died (Fig 4B).

When we tested the virulence of the VBNC subpopulation P1 and P2 independently (separated and sorted by FACS), at time point T12, by injecting approximately $10^4$ CFU (without resuscitation) into the larvae we found that all of the larvae survived and showed no signs of disease after 48 h. Even though death was seen when the microcosm was injected as a whole, larvae death was not observed when subpopulations of VBNC cells were separated and then injected into larvae. SEM analysis of whole microcosms confirms that VBNC cells are embedded in an extracellular matrix (Fig 4A) which is completely removed by the pressure of the rapid flow stream used to separate cells during the FACS process and gives pure defined VBNC subpopulations without any extracellular matrix present (Fig 3B–3D).

## Resuscitated VBNC P2 cells regain virulence

Next, we assessed the virulence of P1 and P2 resuscitated VBNC cells at T12. When approximately $10^4$ CFU of VBNC subpopulation P1 were resuscitated and the cells were injected into larvae, they did not kill larvae within 48 h. Conversely, when approximately $10^4$ CFU of VBNC subpopulation P2 were resuscitated and injected into larvae, all of the larvae died within 24 h (Fig 4B.) This suggests that *V. parahaemolyticus* VBNC cells in subpopulation P2, when resuscitated, revert to virulent forms. Fig 4C shows a heat map of regulation among virulence proteins and indicates no major known virulence mechanisms where upregulated.

## VBNC cells are present in samples of seafood

To determine whether the methods developed to identify VBNC cells *in vitro* were applicable to samples of seafood, we screened a commercial shrimp (*Penaeus vannamei*: king prawn) sample which tested negative for *V. parahaemolyticus* when using the ISO 21872–1 method. The seafood sample was subjected to a temperature increase, by leaving the prawns at room temperature for 12 h, and then processed and incubated at 30°C for 24–48 h. *V. parahaemolyticus* was recovered from the sample demonstrating that increased temperature could resuscitate VBNC cells *in vivo*. Stomached tissues were analysed using FACS and cells were collected using gates corresponding to the P1 and P2 subpopulations that we had seen in our *in vitro* studies (Fig 5A and 5B). Next, we investigated the morphology of cells in the gated regions corresponding to the P1 and P2 subpopulations using IFC. The cells in region P1 consisted of

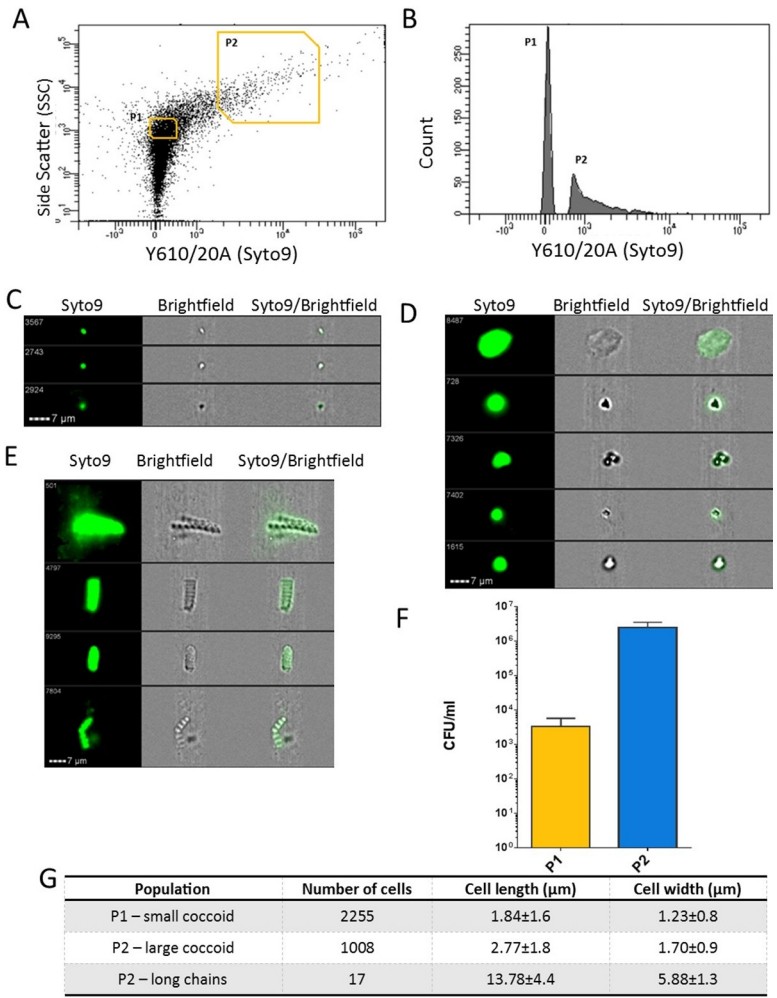

**Fig 5. Analysis of the cell morphology of cells in seafood samples.** Cells were stained with Syto9 stain before imaging. Panel A shows dot plots of side scatter area vs. Y610/20 emission area management using FACS for a seafood sample that was examined. Line gates already determined were used to select population P1 and P2 from the seafood sample and are plotted on a histogram of Y610/20 emission area to highlight proportions of cells in populations P1 and P2 (Panel B). The IFC images in Panel C, D and E show Syto9 stained cells, brightfield and composite images of both syto9 and brightfield together. Examples of small coccoid cells from gated region P1 are shown in Panel C, large coccoid cells and long chains of cells present in gated regions of P2 are shown in Panel D and E respectively. Panel F shows the CFU/ml recovery (resuscitation) of VBNC cells collected from subpopulations P1 and P2. Data is representative of 3 sorts and standard deviation is shown ± SD. Panel G indicates the sizes of cells determined by the IDEAS software.

small coccoid cells that were approximately 1.84 μm in length and 1.23 μm in width (Fig 5C and 5G). The cells in region P2 consisted of large coccoid cells (Fig 5D and 5G) that were approximately 2.8 μm in length and 1.7 μm in size and large chains of rod cells that were approximately 13 μm in length and 5.9 μm in width (Fig 5E and 5G). Using IFC images alone, we cannot be certain that the cells we see are *Vibrio* species however, they do resemble *V. parahaemolyticus* VBNC cells. Cells from P1 and P2 gated areas were collected using FACS and non-resuscitated cells could not be grown on agar while those treated with the resuscitation step were culturable and were identified as *V. parahaemolyticus* (Fig 5F) using ToxR species targeted PCR. These results indicated that this seafood sample contained *V. parahaemolyticus* VBNC cells that were morphologically similar in shape and size to VBNC cells seen in our *in vitro* studies.

**Table 2. Differentially expressed proteins in the chromosomes.**

|  | Chr 1-up[a] | Chr 1 –down[b] | Chr 2- up[a] | Chr 2 –down[b] |
|---|---|---|---|---|
| T0 vs. VBNC state P1-T12 | 70 (2.3%) | 235 (7.6%) | 21 (1.2%) | 70 (4%) |
| T0 vs. VBNC state P2-T12 | 41 (1.3%) | 200 (6.5%) | 10 (0.6%) | 36 (2.1%) |
| T0 vs. VBNC state P1-T50 | 160 (5.2%) | 420 (13.6%) | 37 (2.1%) | 97 (5.5%) |
| T0 vs. VBNC state P2-T50 | 137 (4.4%) | 430 (14%) | 35 (2%) | 105 (6%) |

The differentially expressed proteins were given in total as well as in proportion to the total number of protein-coding genes in each chromosome in brackets.

[a]up, upregulated proteins

[b]down, down regulated proteins; Chr1, chromosome 1; Chr2, chromosome 2.

## Subpopulations of VBNC cells have distinct proteomes

Having established that cells in subpopulation P2 can be resuscitated, the molecular processes involved in VBNC cell formation were investigated. Using quantitative mass spectrophotometry, the proteome of VBNC populations P1 and P2, derived from time points T12 and T50, was resolved and compared to that from cells at T0 time point. We quantified similar numbers of proteins in T0 (n = 1533) or in the P1 or P2 at time points T12 and T50 (n = 1444–1497) (S1 Table). A total of 1690 proteins were detected across all groups, representing 35% of the *V. parahaemolyticus* genome across all the samples, of which 1173 proteins were found in all of the samples (S1 Data).

Regression analysis was used to compare the different proteomes. We found that the P2 proteome was more similar to the T0 proteome than to the P1 proteome at time points T12 and T50 (see S2 Table). We also found that the proteomes of P1 and P2 cells were different from each other at both time points (S2 Table). A comparison of these datasets using principle component analysis of the global expression profiles, confirmed the relationships of the different proteomes to each other (S2 Fig).

Comparisons of the proteome between populations P1 and P2 showed few differences at time points T12 or T50. At time point T12 a total of 4 proteins were significantly downregulated in P1 compared to P2. A total of 27 and 10 proteins were significantly upregulated or significantly downregulated respectively in population P1 at time point T50 compared to population P2 at the same time point. To assess the differences between the proteome populations, we determined the number of proteins that were significantly up and down regulated between the different VBNC populations compared to T0 (Table 2). Proteins with a 3-fold change in level of expression and a q value <0.01 were defined as significantly regulated (Table 2, S2 Data) and assigned into 23 functional groups (Fig 6). The majority of differentially regulated proteins were assigned to metabolism functional categories, for VBNC subpopulations P1 and P2 at time points T12 and T50. More of these proteins were significantly regulated at time point T50, compared to the earlier time point of T12 (Figs 6A and 6B and S2). A collective total of 101 metabolism-associated proteins were upregulated at time point T50 from VBNC subpopulations P1 and P2 (Fig 6B). By contrast, at time point T12, 42 metabolism-associated proteins were upregulated from combined P1 and P2 populations (Fig 6A). Conversely, more metabolism associated proteins were downregulated at T50 than at time point T12 (278 and 152 respectively; Fig 6A and 6B). In the metabolism-related functional category, 'energy production and conversion', there were 15 more upregulated proteins at time point T50 than at time point T12 (28 versus 13). Overall, we observed more proteins downregulated indicating that as the VBNC state progresses from T12 to T50 more proteins are made redundant.

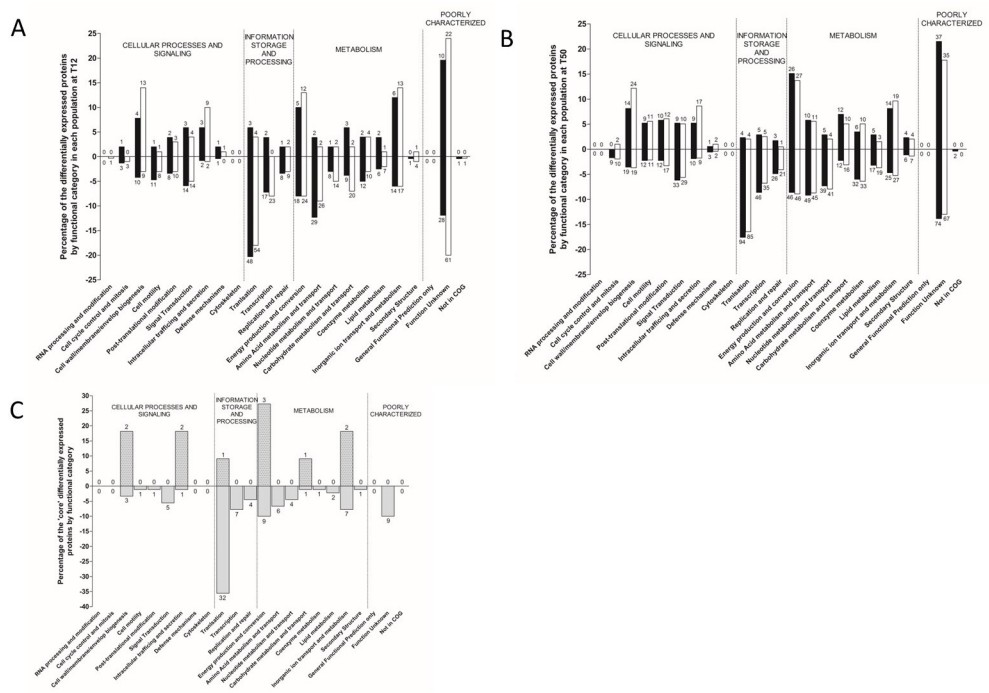

**Fig 6. Functional categories of differentially expressed proteins of *V. parahaemolyticus* cells in the different VBNC subpopulations at each time point.** Only significant expressed proteins (q value < 0.01) with expressional changes of 3 or greater in VBNC subpopulation versus T0 are shown. Bars indicate the portion of the differentially expressed genes by functional category in each population (100% is the number of regulated proteins in subpopulations P1 or P2). The number of proteins in each category appears above the median (upregulated) or below the median bar (downregulated). **A:** shows proteins regulated at T12 time point **B:** shows proteins regulated at T50 time point and **C:** shows core proteins that are shared in subpopulations P1 and P2 at both time points T12 and T50. Subpopulation P1 is in solid white bars while subpopulation P2 is in solid black bars.

Some proteins were differentially expressed (significantly upregulated or significantly down regulated) in subpopulations P1 and P2 at time points T12 and T50 (Fig 6C) and may be core proteins which play a central role in the VBNC state (S3 Fig). From this, we identified 11 proteins that were upregulated (Fig 7) and that clustered together during hierarchical clustering (S3 Fig). These proteins were VPA0166 and VP1267 involved in cell wall membrane, VP0622, VP0589 (YajC) involved in intracellular trafficking and secretion, VP2817 (HfQ) assigned to translation category, VPA1499 (LldD), VP1161 and VP1053 involved in energy production and conversion, VP0240 involved in carbohydrate metabolism and VP0171 and VP0174 involved in inorganic ion transport and metabolism. We identified another 90 proteins that were downregulated in subpopulations P1 and P2 from both T12 and T50 time points. Of these 43 were assigned to the information storage and processing categories, 31 were assigned to metabolism (Fig 6C), 11 were assigned to cellular processing and signalling and 9 could not be characterised (Fig 6C).

## Lactate dehydrogenase promotes resuscitation

A lactate dehydrogenase (LldD; VPA1499) was one of the most highly upregulated proteins identified in subpopulations P1 and P2 at both time points T12 and T50 (Fig 7A–7D). To investigate its role in more detail, strain RIMD2210633:Δ*lldD* was created in which *lldD* was deleted. The RIMD2210633:ΔlldD was confimed by whole genome sequencing, assays assessing growth (data not shown) and virulence (S5 Fig) were checked in *G. mellonella* and showed

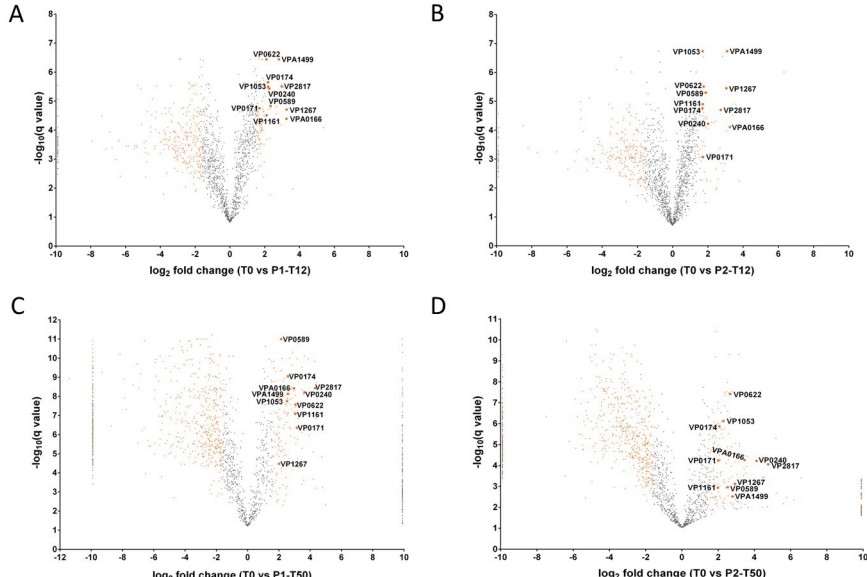

**Fig 7. Quantitative cellular proteomics identifies proteins involved in VBNC state.** Using quantitative mass spectrophotometry, the proteome of VBNC populations P1 and P2, derived from time points T12 and T50, was resolved and compared to that from cells at T0 time point. Volcano plots summarising the proteomic comparison of total proteins between VBNC subpopulations P1 and P2 at time point T12 and T50 in *V. parahaemolyticus*. The x-axis shows the $\log_2$ of the fold change of protein expression plotted against the $-\log_{10}$ of the q value. Orange dots indicate the differentially expressed genes with at least 3-fold change ($\log_2$ 1.585) and statistical significance (q value < 0.01 [$-\log_{10}$ 2]). The dots at the -9.9 and 9.9 value on the x-axis (downregulated and upregulated, respectively) represent proteins that were not detected in the VBNC sample or in the T0 sample, respectively in the pairwise comparison (fold change values of -9.9 and 9.9 were added empirically). Distribution of the differential expression of proteins in VBNC cells in Panel A P1-T12, Panel B P2-T12, and Panel C P1-T50 and Panel D P2-T50 population. Highlighted dots represent the 11 proteins that were significantly upregulated in VBNC subpopulations P1 and P2.

no significant diffrences. At T0 time point the cell of Δ*lldD* (1.899±0.56 μm) was larger in length size compared to T0 timepoint of the wildtype RIMD2210633 (1.26±0.1 μm). When microcosms were established with RIMD2210633:Δ*lldD* the population entered a VBNC state significantly earlier (17 days earlier; P value <0.0001) than wild type RIMD2210633 (Fig 8A). Addition of sodium pyruvate to the microcosms was able to restore the rate of entry into the VBNC state of the RIMD2210633:Δ*lldD* to that of the wild type (Fig 8B). LldD is responsible for converting L-lactate to pyruvate (Fig 8D and 8E). When the RIMD2210633:Δ*lldD* was put into the VBNC state, it showed at timepoint T12 that the morphology of VBNC cells had changed. Overall there were less numbers of large coccoid cells in the P2 population (2%) compared to wildtype RIMD2210633 strain (10%). In the RIMD2210633:Δ*lldD* the large coccoid cells present in the P2 population were also smaller in size (length 3.16±0.15 μm width 1.76 ±0.2 μm) compared to wildtype RIMD2201633 large coccoid cells (length 6.34 ± 1.10 μm, width 4.30 ± 0.48 μm). Also in the P2 population of the RIMD2210633:Δ*lldD*, there were larger number of filaments or chains of cells present (S6 Fig).

An alternative lactate dehydrogenase in *V. parahaemolyticus* RIMD2210633 (VPA0147) can also convert L-lactate to pyruvate, but this protein was not expressed in our VBNC cells. We investigated whether the addition of sodium lactate, sodium pyruvate and sodium acetate would enable the resuscitation of mid-to late stage VBNC cells, when thermal shift methods alone were not sufficient. The addition of 2 mM of sodium lactate resuscitated mid VBNC stage cells (36 days after cells had become unculturable in the microcosm) enabling some cells to be cultured, whilst control VBNC cells incubated in PBS were not resuscitated (Fig 8C). The

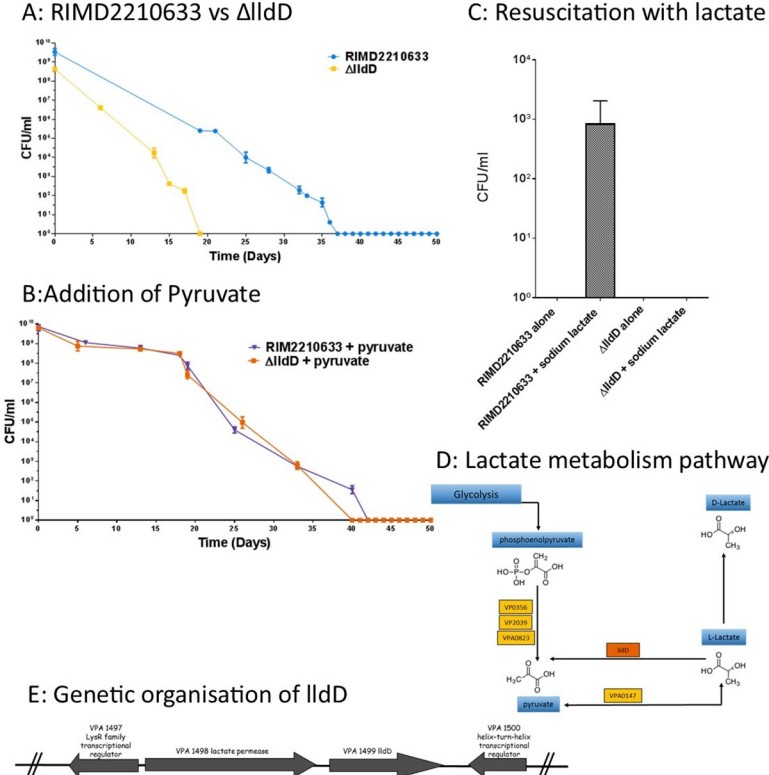

**Fig 8. Resuscitation of dormant cells using sodium lactate.** Panel A shows entry into VBNC state for RIMD2210633:
*ΔlldD* compared to wildtype RIMD2210633. Panel B shows entry into VBNC state for RIMD2210633 and
RIMD2210633:*ΔlldD* in the presence of sodium pyruvate to the microcosm. Panel C shows VBNC cells of
RIMD2210633 resuscitated with 2mM sodium lactate 37 days after entering VBNC stage (mid to late VBNC stage).
Panel D shows the lactate metabolism pathway in *V. parahaemolyticus* RID2210633. Each rectangle stands for an
enzyme in the pyruvate metabolism pathway. The yellow boxes indicate the ortholog genes in *V. parahaemolyticus*
RIMD2210633 that are not present as proteins in the VBNC subpopulations while the green boxes are significantly
downregulated VBNC proteins. The proteins in the orange box indicates genes that are upregulated in VBNC
subpopulations. Panel E shows the genetic organisation for VPA1499 (*lldD*) in *V. parahaemolyticus* RIMD2210633.

addition of other oxidative compunds including sodium pyruvate and sodium acetate did not
resuscitate RIMD2210633 VBNC cells.

When we investigated whether the addition of sodium lactate, sodium pyruvate and sodium
acetate would resuscitate RIMD2210633:*ΔlldD* VBNC cells in mid-late stage we found that
none of these compounds were able to turn the VBNC cells to culturable forms as seen with
the wildtype.

## Discussion

Many bacteria are reported to adopt a survival strategy by entering a VBNC or dormancy
state, when exposed to stressful or non-permissible growth conditions [1,12,32]. Factors
known to induce VBNC formation include nutrient starvation, extreme temperatures, expo-
sure to UV light and chlorination of waste-water treatments. In this study, the VBNC state was
induced in *V. parahaemolyticus* by nutrient restriction and by lowering the environmental
temperature to mimic conditions that occur in the environment during non-permissible
growth conditions. The lag period before all *V. parahaemolyticus* cells become VBNC in a
microcosm has previously been shown to differ depending on the conditions used to set it up.
In some cases the lag period has been reported to be 9–15 days [11,18,27] while Wong *et al.*

showed that environmental strains of *V. parahaemolyticus* took between 35–49 days to become VBNC [16]. Jiang *et al*., found it took 50–80 days for all cells to become VBNC when high salt concentrations were used in the starvation media [8]. In this study, when fresh cultures (<5 days from culturing from feezer stocks) were used to prepare microcosms, we found that it took *V. parahaemolyticus* RIMD2210633 cells approximately 30–35 days to become VBNC. When older seed cultures were used to establish the microcosms it took 20 days for all of the population to become unculturable. For the present studies we have consistently used bacteria that had been cultured <5 days from freezer stocks. Our findings, that the age of the culture can influence the kinetics of formation and the resuscitation of VBNC cells mirrors some of the findings with the formation of persister cells, where the culture conditions affects the frequency of persister cells [33]. These findings resolve some of the differences in VBNC formation reported in the literature.

Bacteria in the VBNC state are viable, or metabolically active, but are unable to form colonies on standard culture media. Following stimuli, such as a temperature upshift, some VBNC cells can 'resuscitate' restoring their abilities to grow. There is much debate about the most appropriate method for determining the numbers of VBNC cells in a population and many studies [5,7,10,13,14,34] measure resuscitation on a rich medium. However, some VBNC cells may not be recovered using this protocol. As *V. parahaemolyticus* is unable to grow in PBS, we used this medium and a temperature upshift to measure the number of cells in the population that could be resuscitated from the VBNC state. This indicated that only 1.3% of cells in an unculturable population were VBNC. These findings are broadly similar to those reported by Bamford *et al*., 2017 who used microfluidics to show that approximately 1% of an antibiotic-treated population of *E. coli* formed VBNC cells [35]. In our studies, the resuscitation window lasted approximately 13 days after all the cells in the microcosm had turned unculturable which is consistent with published literature [15–17].

## Do VBNC cells have the potential to be virulent?

There are some reports that human pathogenic bacteria retain the ability to cause disease in the VBNC state [3,12,36]. For example, one study showed that VBNC cells of *E. coli* O157:H7 continued to produce Shiga-like toxins [9]. Other studies showed that *V. cholerae* VBNC cells are virulent in animal models of disease, and VBNC cells may be linked to seasonal epidemics of cholera [2,4]. The detection of *V. parahaemolyticus* in the environment, and cases of disease in humans, typically peaks in the summer months when warmer sea temperatures allows bacterial proliferation. However, the pathogen is present at low levels, or undetectable (using classical techniques) in environmental samples taken during the winter months [18]. A recent study showed that oysters harvested during the winter months contained *Vibrio* VBNC cells [37]. *V. parahaemolyticus* VBNC cells have also been converted to the culturable form when co-cultured with eukaryotic cells such as HT-29 or Caco-2 cells [38], indicating the potential for *in vivo* resuscitation. Due to the possibility that VBNC cells of pathogenic bacteria can retain their virulence, VBNC cells are a major public health concern in particular in food microbiological safety. The microbial contamination of a food sample is determined by plate count methods and if VBNC cells are present then they could go undetected during routine food microbiology testing due their inherent unculturability. This can lead to an underestimation of the disease potential of that sample. In this study, we found conclusive evidence that *V. parahaemolyticus* RIMD2210633 VBNC subpopulations cannot be resuscitated inside *G. mellonella*. This is supported by the lack of regulation of known virulence proteins including, T3SS1 (VP1658-VP1702), pathogenicity islands (PAIs) PAI-1 (VP0380-0403), PAI-2 (VP0634-0643), PAI-3 (VP1071-1095), PAI-4 (VP2131-2144), PAI-5 (VP2900-2910), PAI-6

(VPA1253-1270), PAI-7 (VPA1312-VPA1395), TDH, capsular polysaccharide (CPS) proteins (VPA1403-1412), lipopolysaccharide (LPS) proteins (VP0218-VP0234), T6SS1 (VP1386-VP1414) and T6SS2 (VPA1025-VPA1046), and we found only eight proteins significantly upregulated (4% of known virulence related proteins) (Fig 4C). If VBNC cells cannot be resuscitated inside the host under these conditions as shown in this study, then the pathogenic nature of the *V. parahaemolyticus* would appear to be curtailed. However, the risk to the consumer would be if there was a lag period or a change in conditions, which then allowed the VBNC cells in the food product time to resuscitate, thus restoring pathogenesis.

So what could account for the death seen in larvae when whole microcosms where injected? It is possible that the death seen in larvae when injected with whole microcosm or with supernatant samples may have been due to a virulence compound present which does not appear in VBNC cell populations collected after FACS. We observe the presence of extracellular matrix that holds VBNC cells in biofilms that are visible in SEM images (Fig 4A). During FACS the microcosm is stained with SYTO9 and the extracellular matrix is not detected on FACS plots and/or is flushed out due to the pressure of the rapid flow stream and thus would not be collected unless it was tightly bound to VBNC cells. This extracellular matrix may be in abundance when cells are injected into larvae without prior separation by FACS resulting in toxicity and the death of larvae recorded. A study by Sarkisova and co-workers [39] showed that the extracellular matrix of *Pseudomonas aeruginosa* in biofilms consisted primarily of the virulence factor alginate and other extracellular proteases that played a role in virulence. Notably, we found a protein VP2692 that is involved in alginic acid biosynthetic process to be upregulated at time point T12 (3.80 fold).

## Formation of distinct VBNC subpopulations

To the best of our knowledge, we report unequivocally that different populations of VBNC cells exist (under the VBNC induction conditions tested in this study) and this is the first report of the quantification of these different VBNC cell types and their different resuscitation potentials. Most studies with VBNC cells in other Gram negative bacteria report cell dwarfing and/or rounding when in the VBNC state [4,6,8,18,40–44]. This study demonstrates that 90% of *V. parahaemolyticus* cells change from rod shaped to small coccoid shaped bacteria when they enter the VBNC state. These cells could be resuscitated for up to 14 days after the population has become VBNC. We also show that 10% of the *V. parahaemolyticus* VBNC population forms large coccoid cells. There was a longer window of time (up to 50 days) during which these cells could be resuscitated. These large coccoid cells retained metabolic activity and once resuscitated again were virulent in *G. mellonella*. Coutard *et al.*, also observed a heterogeneous population of *V. parahaemolyticus* VBNC cells in their microcosms using SEM where there were small coccoid cells as well as flattened larger cells [18]. Importantly, we have also demonstrated using the IFC and FACS approach that large coccoid VBNCs exist in a seafood sample that tested negative for *V. parahaemolyticus* by conventional techniques.

## Lactate dehydrogenase promotes VBNC resuscitation

A number of previous studies have demonstrated that pyruvate, through its properties as an antioxidant, can assist in the VBNC resuscitation process [45,46]. Pyruvate can detoxify the effects of $H_2O_2$, hydroxyl radicals and lipid peroxidation; it can also remove oxygen radicals via a non-enzymatic oxidative decarboxylation reaction to produce water and acetic acid [47,48]. Pyruvate is a key intermediate in central metabolism involved in the TCA cycle, fatty acid synthesis, and the biosynthesis of amino acids which feed into gluconeogenesis.

Recently, the role of pyruvate sensing and transport during resuscitation of *E. coli* VBNC cells following periods of extended cold stress has been reported [45]. Proteomic analysis of *E. coli* VBNC cells have revealed that several enzymes involved in pyruvate metabolism are significantly upregulated, these include the pyruvate formate lyase (PflA), phosphoenolpyruvate carboxykinase (PckA) and L-lactate dehydrogenase (LldD) [45]. Cellular proteomics of *V. parahaemolyticus* VBNC cells combined with genetic analysis led to the discovery of L-lactate dehydrogenase (LldD:VPA1499) being significantly up-regulated in VBNC cells. L-lactate dehydrogenase catalyses the oxidation of lactate to pyruvate (Fig 8D) and feeds electrons back into the electron transport chain, thereby fueling cellular respiration. Our results show that deletion of LldD created a strain that when, in microcosms, entered the VBNC state significantly quicker than the WT strain. It was also shown that supplementation with lactate aided resuscitation and extended the resuscitation window. Furthermore the addition of pyruvate (the product of lactate oxidation) to microcosms of the *ΔLldD* strain restored the WT characteristics and reduced the rate of VBNC formation.

The proteomic analysis also identified the regulatory protein Hfq (VP2187) as a significantly up-regulated protein in the *V. parahaemolyticus* VBNC population. Hfq in *V. parahaemolyticus* has been shown to down regulate catalase (CAT: VPA0453) and superoxide dismutase (SOD: VP2118) [49], both of which aid VBNC resuscitation and play a key role in the oxidative stress response. Our analysis also shows that CAT and SOD are down-regulated in both the VBNC subpopulations. Interestingly, Hfq is more highly up-regulated in P1 cells than in P2 (P1-T12: 7.87 fold change and P2-T12: 6.64 fold change); concomitant with both CAT and SOD being more down regulated in the P1 subpopulation. The combined analysis from our biochemical and proteomic studies suggest that *V. parahaemolyticus* VBNC resuscitation potential is (at least in part) dependent on the ability to combat oxidative stress.

In summary, we have distinguished two key subpopulations (P1 and P2) of *V. parahaemolyticus* VBNC cells that are able to stay dormant in the VBNC state for long periods of time. It has been demonstrated that these different subpopulations of VBNC cells display distinctive abilities for revival under favourable conditions and through proteomic analysis have identified a number of key proteins expressed in populations P1 and P2 at different time points that may play an important role in VBNC formation and resuscitation.

## Methods and materials

### Bacterial strains and cultures conditions

Bacterial strains used in this study are shown in Table 3. *V. parahaemolyticus* strains were initially cultured aerobically onto selective media Thiosulphate Citrate Bile Sucrose (TCBS) agar (Oxoid) at 37˚C for 24 h. For enumeration of colony counts, routine sub culturing and growth on Marine Agar (Conda Labs, Spain) was used and incubated at 30˚C for 24 h.

**Table 3. Bacterial strains and plasmids used in this study.**

| Bacterial Strain | Description | Source or Reference |
|---|---|---|
| *V. parahaemolyticus strains* | | |
| RIMD2210633 | Clinical isolate | [50] |
| RIMD2210633:*ΔlldD* | lldD mutant | This study |
| *E. coli strains* | | |
| DH5*α* Δpir | *recA1 gyrA* (Nal) Δ(*lacIZYA-argF*) (φ80d*lac*Δ[*lacZ*]M15) *pirRK6* | [51] |
| pKR2013 (helper strain) | plasmid mediates conjugation-based plasmid transfer; Kanamycin resistance | [52] |
| Plasmids | | |
| *pDM4* | Suicide vector with R6K origin: Chloramphenicol resistance | [53] |

## Microcosm assay for preparation of VBNC cells

To prepare *V. parahaemolyticus* high density microcosms, *V. parahaemolyticus* strain RIMD2210633 was grown overnight in Marine Broth at 37˚C. The following day 50 ml of fresh Marine Broth was incubated with the overnight culture and allowed to grow until an $OD_{595}$ of 1.3 was reached. The culture was centrifuged at 15000 x g for 10 mins at 6–8˚C. The cells were washed in 40 ml modified PBS solution (0.4 g/L NaCl, 0.1 g/L KCl, 1.45 g/L $Na_2HPO_4$, 0.1 g/L $KH_2PO_4$) and finally re-suspended in 40 ml of modified PBS. The prepared microcosm was placed in the fridge (6–8˚C) until required. Plate counts of culturable cells were carried out periodically over subsequent days until the microcosm had no culturable cells visible on Marine agar. Three time points are described in this work, firstly, T0 on day 0 when cells were first set up into the microcosm, secondly, time point T12 which was taken 12 days after the whole microcosm had turn unculturable and thirdly, time point T50 which was taken 50 days after the whole microcosm had turned unculturable.

## Resuscitation of VBNC cells

Basic resuscitation: When no culturable cells were detected in the microcosm, basic resuscitation of VBNC cells was carried out by placing 500 µl of the microcosm into 4.5 ml of PBS for 5–6 h at room temperature and then incubated at 30˚C overnight (18 h). Cell counts of VBNC cells in the microcosm were carried out on Marine agar. Resuscitation using 2 mM sodium pyruvate, 2 mM Sodium D-Lactate or 2 mM Sodium acetate was carried by centrifuging 2–5 ml of microcosm at 13000 rpm. Collected cells were then re-suspended in pyruvate/lactate/acetate containing medium and left at room temperature for 5–6 h and then incubated at 30˚C for 36–48 h before cells were plated out onto Marine agar and TCBS agar for counts.

## Fluorescent activated cell sorting (FACS) to separate VBNC cells

For flow cytometry analysis, 1 ml of microcosm was stained with 2 µl of Syto9/Propidium Iodide (PI) mix (equal volumes) (live/dead staining for viability) and left in the dark for 15 mins. Fluorescence of prepared samples was measured using a BD FACS Aria III (Becton Dickenson, USA) equipped with a 100 µm nozzle. Particle forward scatter and side scatter were obtained using a 488 nm laser and 488 ± 10 nm detector. Particle fluorescence was measured at 488 nm excitation, 530 ± 30 nm emission for the PI stain, and 561 nm excitation, 610 ± 20 nm emission for the Syto9 stain. Control experiments using log phase bacteria were used to focus population gates around *V. parahaemolyticus* cells that were alive and had an intact cell membrane (using Syto9 stain) (S4 Fig). Boiled bacterial suspensions were used to focus population gates around dead/damaged cells (using propidium iodide (PI) stain) (S4 Fig). Approximately 50–250,000 cells/events of different subpopulations were collected into PBS. The FACS collected cells were centrifuged for 20 min at 4000 rpm and re-suspended in 1 ml of PBS. They were left at room temperature for 5–6 h and then placed at 30˚C for 48 h to allow resuscitation. Cell counts of VBNC cells in the microcosm were carried out on Marine agar.

## Analysis of imaging flow cytometry (IFC) data on VBNC cells

Imaging flow cytometry was applied to characterise the morphology of cells in the identified subpopulations. A total of 10,000 cells from each subpopulation were sorted by FACS into mPBS. Cells were centrifuged at 13000 rpm and re-suspended in 500 µl of mPBS containing Syto9 stain and left for 15 min. The stained cells were then centrifuged at 13,000 RPM for 5 min and re-suspended in 500 µl of 4% paraformaldehyde (PFA) and left for a further 15 min to be fixed. The cells were then washed twice in PBS and stored at 4˚C until required.

IFC data acquisition was performed using a fully calibrated (ASSIST tool) ImageStream X MkII (ISX, Luminex Corp, Seattle, USA) configured with a single camera and 405, 488, 642 and 785 nm excitation lasers, brightfield illumination and a six channel detection system. For maximum resolution and high sensitivity, fluidics were set at low speed, magnification was set at 60x (0.3 $\mu m^2$/pixel) and excitation lasers were set accordingly: 488 nm (100 mW) and SSC (785 nm) at 10 mW.

A minimum of 2000 in-focus single cell events were collected for each sample. Only data from relevant channels were analysed including Channel 02 (CH02 for Syto9 Green detection 533/55 nm), Channel 04 (CH04 brightfield (BF) 610/30 nm) and Channel 06 (CH06, side scatter (SSC) 762/35 nm). To adjust for spectral overlap between these channels, a compensation matrix was applied, calculated from data acquired excluding BF and SSC laser excitation.

Analysis of IFC data was achieved using the IDEAS software (Version 6.2, EMD Millipore, Seattle, USA). Firstly, a scatter plot of fluorescence intensity of Channel 02/Channel 06 was used to exclude background material, and image captures of multiple cells. The default 'mask' for each channel, a region superimposed over channel images used for displaying feature-value calculations, was refined using the adaptive erode, intensity and raw max pixel mask tools. This enabled a more accurately defined Brightfield (BF) image and fluorescence signal from which quantitative morphological and intensity data were derived.

Next, a gradient RMS (root mean square for image sharpness) histogram was used to exclude unfocused cells. Subsequently, the average size of focused cells was determined using length (BF or fluorescence) and width (BF or fluorescence) histograms.

## Scanning electron microscopy (SEM) of VBNC cells

For SEM, we analysed healthy log phase bacteria of RIMD2210633, cells when they were just set up into the microcosm (T0) and FACS sorted VBNC cells from two subpopulations. Cells suspended in medium were fixed in 2% glutaraldehyde and 2% paraformaldehyde in 0.1 M sodium cacodylate buffer pH 7.2 for 1 h at room temperature and could be stored in fixative at 4˚C until further processing. Cells were subsequently washed 3 x 5 min in buffer then post-fixed in 1% aqueous osmium tetroxide for 1 h. After 3 x 5 min washes in deionized water cells were dehydrated through a graded ethanol series (30, 50, 70, 80, 90, 95% ethanol for 5 min per step then followed by 2 x 10 min in 100% ethanol). While suspended in 100% ethanol cells were passed through a 0.22 micron polycarbonate filter (Osmonics Inc., Lenntech, Delfgauw, The Netherlands) using a mild vacuum and the filter instantly suspended in ethanol again. Cells were then fully dehydrated in hexamethyldisilazane (HMDS, Merck, Southampton, UK) for 3 min followed by air drying. Alternatively, cells were passed through a 0.1 micron polycarbonate filter (Whatman nucleopore track-etch membrane, Merck, Southampton, UK) directly following the initial fixation and then processed as described above.

The filter with the dehydrated cells was then mounted on an aluminium stubs and coated with a 10 nm layer of gold/palladium (80/20) using a Q150TES sputter coater (Quorum Technologies Ltd, Laughton, UK) and could be imaged using a GeminiSEM 500 scanning electron microscope (Carl Zeiss Ltd, Cambridge, UK) operated at 5 kV.

## Mass spectrophotometry analysis of *V. parahaemolyticus* VBNC proteome

For proteomics, we used FACS to sort, collect and isolate protein from *V. parahaemolyticus* VBNC cells from subpopulations P1 and P2 at two different time points; T12 and T50. Collected cells were immediately centrifuged and resuspended in BugBustergee Protein Extraction reagent (Novagen, Merck and Co. Inc) to lyse the bacterial cells and release proteins. Control extractions of bacterial proteins were carried out on Day 0 (T0) when bacterial cells were first

prepared in microcosms and had not been subjected to cold stress. Mass spectrometric analysis of protein samples from either 2 or 3 biological replicates was done by the University of Bristol Proteomics Facility. Proteomics was performed as described previously using an UltiMate™ 3000 nano HPLC system in line with an LTQ-Orbitrap Velos mass spectrometer (Thermo Scientific) [54]. The raw data files were processed and quantified using Proteome Discoverer software v1.2 (Thermo Scientific) and searched against *V. parahaemolyticus* RIMD2210633 RAST ORFs using the SEQUEST algorithm. The reverse database search option was enabled and all peptide data was filtered to satisfy false discovery rate (FDR) of 5%. Abundance of each protein in each sample was calculated using the average area measurements of the three most abundant peptides matching to each protein (Top3 method) [55]. This value was then expressed in the fraction of the signal derived from all the proteins detected in each sample. Comparisons were then made for each protein detected in the different time points and population types. Statistical significance of the fold change difference was calculated by performing Bayes moderated t test using R programming. All proteins with a q-value <0.01 and at least 3-fold change difference in expression were considered significantly regulated.

## Online tools

Cellular localisation of the proteins encoded in the *V. parahaemolyticus* RIMD2210633 genome was predicted using PSORTb v.3.0.2 (https://www.psort.org/psortb/) [56]. *V. parahaemolyticus* RIMD2210633 proteins were classified into functional categories based on clusters of orthologous gene (COG) designations; COG categories were assigned to each protein using eggNOG-mapper (http://eggnog-mapper.embl.de/) [57,58]. Global proteomes were compared by principal component analysis (PCA). Heatmap and hierarchial clustering of the (significantly) expressed proteins were generated using Heatmapper (http://heatmapper.ca/expression/) [59].

## lldD mutant construction

DNA fragments (500bp) including upstream and downstream regions of *lldD* and flanked by SphI and SpeI restriction enzymes were created using GeneArt Gene Synthesis services (ThermoFisher Scientific). The DNA fragment was cloned into plasmid pDM4 via the SphI and SpeI sites. The presence of the cloned DNA was confirmed by PCR using primers 5'-CAGGTAACATGATTGCCATTCACAACG-3' and 5'-ATCTCAAGCAAGTGTGAGAGTGTATTGG-3'. The plasmid pDM4-lldD was maintained in *E. coli* DH5α cells and selected on LB agar containing 50 µg/ml chloramphenicol. For conjugation of *V. parahaemolyticus* RIMD2210633 1 ml of an overnight culture of the recombinant *E. coli* pdm4-lldD (donor strain), *E. coli* pKR2013 (helper strain) and the *V. parahaemolyticus* RIMD2210633 (recipient strain) were centrifuged for 2 min. Supernatants were discarded and the pellets re-suspended in 0.5 ml LB medium. A 100 µl aliquot was spread onto a LB agar plate, either individually or with donor, helper and recipient mixed together (ratio 1:1:4), and incubated overnight at 37˚C. The cells were then re-suspended in 1 ml sterile PBS. Aliquots of 100 µl were plated onto LB agar plates supplemented with 100 µg/ml chloramphenicol and incubated overnight at 37˚C. Colony growth was scraped off using a sterile 10 µl loop and re-suspending in 1 ml of sterile PBS and aliquots of 100 µl were plated onto TCBS plate's supplemented with 50 µg/ml chloramphenicol. After incubation at 37˚C for 5–7 days, colonies were transferred onto fresh LB plates containing chloramphenicol. The transconjugants were grown in LB broth without supplementation overnight, serially diluted in PBS and plated onto salt free LB agar containing 10% (w/v) sucrose. The plates were incubated at 24˚C for 2 to 5 days and colonies screened for chloramphenicol sensitivity and on TCBS agar. In order to confirm that chloramphenicol

sensitive colonies contained the desired mutation, PCR was carried out using *ΔlldD* confirmation primers 5' ACG TAT CTT CAT CAA CTC AGG TGT GAA C-3' and 5'TGACTATGCG CTTGTACATAGTTTTGTAAATC-3'. The genotype of the RIMD2210633:*ΔlldD* was confirmed by genome sequencing using an Illumina HiSeq 2500 platform. Sequence data was aligned against the RIMD2210633 reference genomes using the Illumina GA software. The aligned reads were then visualised using the software program Intergrated Genomincs Viewer (IGV) [60]. Genomic regions with no reads were interpreted as missing from the sequenced genome. The deletion mutant RIMD2210633:*ΔlldD* was used in our subsequent experiments.

## Infection of *Galleria mellonella* larvae

*G. mellonella* larvae called TruLarv were purchased from Biosystems Technology, Exeter, Devon, UK. Larvae weighing between 0.2–0.35 g were chosen for experiments. For each experiment a total of 10 larvae were used per strain to be tested. A total of 50,000 cell/events were collected of cells in the microcosm at both time points T12 and T50 and from population P1 and P2. Cells were resuscitated in PBS with increasing temperature as described previously. Cells were then centrifuged (at 13000 rpm) and resuspended in 100 μl of fresh PBS. The larvae were infected by micro-injection (Hamilton Ltd) into the right foremost proleg with approximately 5000 CFU of *V. parahaemolyticus* in 10 μl volumes. For comparison, 10 μl of cells from the microcosm were directly injected into each larvae, a further 1 ml of the microcosm was centrifuged and 10 μl supernatant was injected into each larvae, while the pellet was re-suspended in fresh PBS and then injected into each larvae. For control purposes, 10 larvae were inoculated with PBS. The larvae were incubated at 37°C and survival was recorded for all strains after 24 h and 48 h. Larvae were scored as dead when they ceased moving, and failed to respond when gently manipulated with a pipette tip. Observation findings were also recorded if larvae colour changed from their normal pale cream coloration to brown or black indicative of melanisation.

## Sample preparation and testing

Frozen samples of *Penaeus vannamei* (shrimp) were received from a UK food supplier and stored at -20°C until testing was performed. Within one month of receipt, the sample was analysed according to ISO 21872–1 with minor modification [19]. Twenty-five grams of prawn meat (taken from a minimum of six prawns) was stomached before the addition of 225 ml of alkaline salt peptone water (ASPW; Oxoid Ltd., Basingstoke, Hampshire, UK). All samples were incubated at 41°C for 6 h after which a 5 μl loopful was taken from directly below the surface of the broth and streaked onto TCBS plates. All TCBS plates were incubated at 37°C for 24 h. Typical sucrose negative (green) colonies were subcultured onto marine agar (Conda labs, Spain) and incubated at 30°C for 24 h. Presumptive colonies were identified as *V. parahaemolyticus* if they met the following criteria; positive for oxidase, negative for Voges Proskauer and Ortho-nitrophenyl-β-D-galactopyranoside, no growth in 0% NaCl and no acid from sucrose. Further identification by API 20E strips (BioMerieux) was also carried out. All biochemically identified *V. parahaemolyticus* strains were further analysed by PCR amplification using the species target *toxR* [61].

After 20 months of storage at -20°C the prawns sample was analysed again using FACS and IFC methods to identify any VBNC *V. parahaemolyticus* cells in the sample. In brief, 25 g grams of prawn meat (taken from a minimum of six prawns) was stomached before the addition of 225ml of PBS was added. Approximately 50–70 ml of the supernatant was filtered through a 100 μm cell strainer to remove excess prawn meat. Of this filtrate, 1 ml was stained with Syto9 stain and IFC data was collected immediately on the sample as described in detail above. Another, 1 ml was stained using Syto9/Propidium Iodide (PI) and using FACS 50,000

events were collected in gated areas for populations P1 and P2 as determined from experiments described above. The FACS collected cells were centrifuged for 20 mins at 4000 rpm and re-suspended in 1ml of PBS. They were left at room temperature for 5–6 h and then placed at 30˚C for 48 h to allow resuscitation. Cell counts of any typical *V. parahaemolyticus* colonies were carried out on TCBS plates and 5–10 colonies were confirmed by PCR amplification using the species target *toxR* [61].

## Supporting information

**S1 Text. Factors to enable VBNC formation.**
(DOCX)

**S1 Fig. Transmission Electron Microscope (TEM) images of VBNC subpopulations.**
(TIF)

**S2 Fig. Principal component analysis of global protein expression profiles of V. parahaemolyticus obtained from bacteria grown in vitro and in VBNC state.** Replicates of the Log Phase cells, T0 and T12 samples are clustered together indicating experimental reproducibility. Replicates of the T50 samples are more distant from each other.
(TIF)

**S3 Fig. Heat map and hierarchical clustering of the significantly upregulated and significantly downregulated proteins in the VBNC populations.** Protein expression data is shown only if the fold change was significantly different in the VBNC populations, compared to protein expression in the P0 population. Heat map shows $\log_2$ fold change difference in protein expression: highly downregulated–dark blue, highly upregulated–yellow. -9.9 or 9.9 $\log_2$ fold change values were assigned empirically to the protein if the protein was not detected in the VBNC population or in the T0 population, respectively. Locus tags of proteins which were significantly upregulated or significantly downregulated in all VBNC populations are highlighted with green or red, respectively.
(PDF)

**S4 Fig. FACS Controls.** Dot plots (Left) and corresponding histograms (Right) of control experiments using boiled bacterial suspensions (A and B) were used to identify *V. parahaemolyticus* cells around dead/damaged that had a comprimsed membrane (using propidium iodide (PI) stain). Dot plots (Left) and corresponding histograms (Right) of control experiments using log phase bacteria (C and D) were used to identify *V. parahaemolyticus* cells that were alive and had an intact cell membrane (using Syto9 stain).
(TIF)

**S5 Fig. *Galleria mellonella* infection with RIMD2201633 and RIMD2210633:Δ*lldD*.** A dose of $10^5$ CFU of RIMD2210633 or RIMD2210633:Δ*lldD* CFU was injected into larvae. Percentage survival was measured after 48 hours. There was no significant difference between virulence of the wildtype and the RIMD2210633:Δ*lld*.
(TIF)

**S6 Fig. Characteristics of the RIMD2210633:Δ*lldD*.** Microcosms of the mutant RIMD2210633:Δ*lldD* were prepared and allowed to enter VBNC state. After 12 days in the VBNC cells were stained with Syto9 and examined for morphology using Imagestream Technology. Panel A and B show cells of the P2 population that were large coccoid or long filaments respectively. Panel C is a table indicating the cell lengths and widths of the cells.
(TIF)

**S1 Table. Protein data and the numbers of proteins detected in each group.**
(DOCX)

**S2 Table. Correlation between the proteomes of the analysed groups.** Determined by regression analysis. Mean of the normalised abundance values were used with each group.
(DOCX)

**S1 Data. Distribution of *V. parahaemolyticus* proteins expressed at T0 and in VBNC subpopulation P1 and P2.**
(XLSX)

**S2 Data. List of significantly upregulated or significantly downregulated proteins in VBNC subpopulations compared to T0.**
(XLSX)

## Acknowledgments

We thank Dr Christian Hacker and Paulina Cherek from the Bioimaging Facility at University of Exeter for performing Scanning Electron Micrograph. We thank Kate Heesom at the University of Bristol Proteomics Facility for carrying out mass spectrophotometry.

## Author Contributions

**Conceptualization:** Sariqa Wagley, Richard W. Titball, Clive S. Butler.

**Data curation:** Sariqa Wagley, Andrea Kovacs-Simon, Zheng R. Yang.

**Formal analysis:** Sariqa Wagley, Andrea Kovacs-Simon, Zheng R. Yang, Ann Power.

**Funding acquisition:** Sariqa Wagley, Richard W. Titball, Clive S. Butler.

**Investigation:** Sariqa Wagley.

**Methodology:** Sariqa Wagley, Helen Morcrette, Andrea Kovacs-Simon.

**Project administration:** Sariqa Wagley, Richard W. Titball, Clive S. Butler.

**Resources:** Sariqa Wagley, Ann Power, Richard K. Tennant, John Love, Neil Murray, Richard W. Titball, Clive S. Butler.

**Software:** Sariqa Wagley, Ann Power.

**Supervision:** Sariqa Wagley, Richard W. Titball, Clive S. Butler.

**Validation:** Sariqa Wagley.

**Visualization:** Sariqa Wagley.

**Writing – original draft:** Sariqa Wagley.

**Writing – review & editing:** Sariqa Wagley, Andrea Kovacs-Simon, Ann Power, Richard K. Tennant, Richard W. Titball, Clive S. Butler.

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
