## [Decision Letter · Decision Letter 0]

25 Sep 2020

Dear Dr Wagley,

Thank you very much for submitting your manuscript "Bacterial dormancy: a subpopulation of viable but non-culturable cells demonstrates better fitness for revival." for consideration at PLOS Pathogens. As with all papers reviewed by the journal, your manuscript was reviewed by members of the editorial board and by several independent reviewers. In light of the reviews (below this email), we would like to invite the resubmission of a significantly-revised version that takes into account the reviewers' comments.

We cannot make any decision about publication until we have seen the revised manuscript and your response to the reviewers' comments. Your revised manuscript is also likely to be sent to reviewers for further evaluation.

Sincerely,

Andreas J Baumler

Associate Editor

PLOS Pathogens

Karla Satchell

Section Editor

PLOS Pathogens

Kasturi Haldar

Editor-in-Chief

PLOS Pathogens

orcid.org/0000-0001-5065-158X

Michael Malim

Editor-in-Chief

PLOS Pathogens

orcid.org/0000-0002-7699-2064

Reviewer's Responses to Questions

**Part I - Summary**

Reviewer #1: The paper submitted by Wagley et al. present interesting findings on VBNC in Vibrio vilnificus (Vv). The authors shows that 2 subpopulation of VBNC cells exits: P1, composed of composed of small coccid cells, and P2, composed of larger, elongated or coccoid cells. The authors further characterize these 2 subpopulation by testing infectivity in Galleria and through proteomic analysis. The identification of these subpopulation is novel and interesting. The proteomic study is very useful and novel. The identification of lldD as a key protein for resuscitation is very interesting and will lead to new studies on the energetics of resuscitation.

Reviewer #2: In the manuscript titled “Bacterial dormancy: a subpopulation of viable but non-culturable

cells demonstrates better fitness for revival.”, Wagley et al., investigated the population

heterogeneity of the viable but non culturable (VBNC) state of V. parahaemolyticus. Using FACS and IFC, they thoroughly and convincingly showed that distinct populations exist with VBNC population. The authors went on to explore the differences between these populations and the starting microcosm using proteomic analysis, showing that lldD was one of the proteins that were differentially regulated. They then showed that lldD played a role in regulating the VBNC state and the metabolic product of lldD was able to rescue the loss of lldD. Overall, the study is carefully designed, compelling and identified new components in regulating bacterial dormancy.

Reviewer #3: In this study, Wagley et al., investigated the mechanism by which different populations of V. parahaemolyticus enter the VBNC state using imaging flow cytometry (IFC), fluorescence activated cell sorting (FACS), and proteomics. The authors identify two distinct bacterial VBNC subpopulations (“P1” and “P2”) based on size and florescence intensity and find that these different abilities for revival. The authors also report altered virulence properties of these subpopulations in a larval model of infection. Proteomic analysis of these subpopulations identified lactate dehydrogenase as a bacterial factor that contributes to the VBNC state and addition of exogenous lactate was able to promote VBNC resuscitation.

Overall, the experiments seem generally well performed and the data are interesting. However, further genetic analysis to investigate the mechanism by which lactate dehydrogenase contributes to VBNC state formation and improved experimental design with their infection model would solidify these findings.

**Part II – Major Issues: Key Experiments Required for Acceptance**

Reviewer #1: 1. For the flow cytometry please present the live and dead (boiled) controls. This would help the reader understand what was done, and how the gates were set. In addition, the result section should be revised. My understanding is that PI was used to test for viability, which was detected by Y610/20. P2 is therefore less viable than P1, but the author do not consider P2 as dead. Again, the controls (live and dead) would help the reader interpret this important dataset. Why not used Syto 9 signal instead of SSC? Was the flow cytometry done on 14 days stored culture or 5 day old culture (in reference to Fig. 1).

2. I am not convinced that the seafood samples contained Vv. I presume that the shrimps harbour a variety of microorganisms that could results in similar images. How does the author confirm that the cells the see are Vv? Microscopic morphology cannot be used to identify microorganisms, unless some kind of specific labeling was used (such as antibodies but this does not seem to be the case).

3. In general, I find that the figures are of poor quality (grainy, font too small) and the legends could be significantly improved. I cannot read protein label on the proteome figures. The authors needs to describe what the experiment is and provide sufficient details for the reader to understand the figure without having to constantly refer to different part of the text.

4. Several section of the results needs to be rewritten. The text is disjointed, the panels are not presented in the same order in the text and in the Figure and some panels are not presented at all (Fig 4B) and (Fig. 8B). It ws very difficult for me to understand what was done. I suggest that the author present the experiments, and the important details, before presenting the results of it. In particular, the microcosm should be presented (conditions, medium, etc) as well as the general method to test resuscitation (PBS). I find the use of T12 and T50 confusing (as it seems to relate to time after the population is composed of VBNC cells only) rather than days. Also Fig 1B present data as VBNC+number of days. I presume VBNC is a specific time point but this is unclear.

5. What is Fig. 4B?

Reviewer #2: 1. Can the author provide possible explanation to why P2 has high PI signal but intact membrane?

2. It is unclear whether the presence of distinct VNBC populations is specific to the condition tested in the manuscript. The authors should test whether distinct VNBC populations are also present when VNBC state is induced by a different stimulus, in order to support their claim (line 484-486).

3. The authors suggested that VBNC subpopulations cannot be resuscitated inside G. mellonella, and that the larvae death was caused by the virulent compounds associated with whole microcosm. A direct way to test this hypothesis will be to infect the larvae with FACS-sorted P1 or P2 population without resuscitation.

4. While it is definitely interesting that the addition of pyruvate can resuscitate VBNC, the underlying mechanisms remains elusive. It will be interesting to test the proposed antioxidant mechanism by assaying whether other antioxidant compounds can rescue the ldhA mutant.

Reviewer #3: -In Figure 4, the authors state that they infected larvae with approximately 10^7 CFU of T0 cells and approximately 10^5 VBNC cells. Later they state that they infected with approximately 10^4 CFU of P1 or P2 sorted cells for infection. It’s difficult to make any conclusions on VBNC virulence given that the authors are using 2-3 logs less VBNC bacteria for infection. The authors should repeat the experiments by normalizing the VBNC cells up to 10^7 or reduce the inoculum of T0 to match the dose being used for VBNC/P1/P2 cells in this figure.

-In Figure 4, the authors state that, “V. parahaemolyticus VBNC cells in subpopulation P2, when resuscitated, revert back to virulence capabilities of the log phase bacteria.” However, the authors do not demonstrate directly that virulence is responsible for this observation and it’s not clear whether increased virulence or if growth alone is responsible for reduced larval survival. The authors should address this by repeating the post-resuscitation P2 infection experiment using a non-virulent bacterial mutant and showing that larvae survival is rescued.

**Part III – Minor Issues: Editorial and Data Presentation Modifications**

Reviewer #1: Line 27: "displayed different abilities", please be more explicit. Which one is better?

Author summary should be revised. It reads like it was written at the last minute, without revision.

Line 53-65: the first paragraph of the intro needs reference for each statement.

L230: different number of "event" were sorted from P1 and P2, could that affect the results of the infection assay?

L242 and 249: I guess the author means Fig 4A, not Fig 4?

L 259-263: this was hard to understand, the text should be revised. I don't see the link with the other part of this paragraph.

L337: what does "more protein made redundant" means?

Fig 6C is unclear. Are does proteins shared between P1 and P2?

Fig 8B: the authors always show the curve until the population becomes all VBNC. Why not do the same for this?

L424-425: "For the present studies we have consistently used fresh cultured bacteria." is in contradiction with the methods for the microcosm that mentioned 5 day old culture and 14 days stored culture.

L468: Fig 4B?

Reviewer #2: 1. In the intro, the transition between using transcriptomic and proteomic is not well supported. If transcriptomics cannot identify unannotated genes that are important for VBNC, it is unlikely that proteomics can.

2. The low image quality of multiple figures significantly impacts the readability of the manuscript. Some figures were unreadable.

3. The authors need to provide experimental details as what were considered as “stained as live and metabolically active” in the main text.

4. Can the author provide possible explanation why P2 has high PI signal but intact membrane?

5. How did the author determine the bacteria in the seafood were V. parahaemolyticus?

6. What percentage of V. parahaemolyticus population in the seafood samples were VBNC? If all detected V. parahaemolyticus were VBNC cells, and these cells cannot be resuscitated in the host, does entering VBNC state mean a pathogenesis dead-end for V. parahaemolyticus? The authors should expand a bit on this in the discussion.

Reviewer #3: -In Figure 1, I’m surprised by the low amount of variation (i.e small error bars) and apparent high level of reproducibility given the complexity of these experiments. Given the number of unknown factors that may contribute to the findings reported in Fig. 1, the authors should state explicitly how many biological replicates were performed to produce these data (this is not mentioned in the Results or Methods). If it wasn't already done, these experiments should be repeated independently at least 3 times to solidify the conclusions made in Figure 1, which sets the groundwork for the rest of the manuscript.

-In Figure 3, the images should be labeled within the figure so that the reader doesn’t have to rely exclusively on the figure legend.

-It would be interesting to analyze the cell morphology of the ΔlldD mutant

-In Figure 8C, the authors should repeat this experiment (addition of sodium lactate) using the ΔlldD mutant.

-Does inactivation of TCA cycle metabolism gene(s) also lead to cells entering VBNC state significantly earlier (like ΔlldD mutant)? Can you resuscitate VBNC stage of cells with other metabolites such as glycerol?

-Does a ΔlldD mutant have altered virulence in the larval infection model? The authors should address this question experimentally.

PLOS authors have the option to publish the peer review history of their article (what does this mean?). If published, this will include your full peer review and any attached files.

Reviewer #1: No

Reviewer #2: No

Reviewer #3: No
---

## [Editor Report · Decision Letter 1]

27 Nov 2020

Dear Dr Wagley,

We are pleased to inform you that your manuscript 'Bacterial dormancy: a subpopulation of viable but non-culturable cells demonstrates better fitness for revival.' has been provisionally accepted for publication in PLOS Pathogens.

Best regards,

Andreas J Baumler

Associate Editor

PLOS Pathogens

Karla Satchell

Section Editor

PLOS Pathogens

Kasturi Haldar

Editor-in-Chief

PLOS Pathogens

orcid.org/0000-0001-5065-158X

Michael Malim

Editor-in-Chief

PLOS Pathogens

orcid.org/0000-0002-7699-2064
---

## [Editor Report · Acceptance letter]

5 Jan 2021

Dear Dr Wagley,

We are delighted to inform you that your manuscript, "Bacterial dormancy: a subpopulation of viable but non-culturable cells demonstrates better fitness for revival.," has been formally accepted for publication in PLOS Pathogens.

Best regards,

Kasturi Haldar

Editor-in-Chief

PLOS Pathogens

orcid.org/0000-0001-5065-158X

Michael Malim

Editor-in-Chief

PLOS Pathogens

orcid.org/0000-0002-7699-2064